# No Impact of Seasonality of Diagnoses on Baseline Tumor Immune Infiltration, Response to Treatment, and Prognosis in BC Patients Treated with NAC

**DOI:** 10.3390/cancers14133080

**Published:** 2022-06-23

**Authors:** Beatriz Grandal, Ashwaq Aljehani, Elise Dumas, Eric Daoud, Floriane Jochum, Paul Gougis, Judicaël Hotton, Amélie Lemoine, Sophie Michel, Enora Laas, Marick Laé, Jean-Yves Pierga, Khaoula Alaoui Ismaili, Florence Lerebours, Fabien Reyal, Anne Sophie Hamy

**Affiliations:** 1Residual Tumor & Response to Treatment Laboratory, RT2Lab, Translational Research Department, INSERM, U932 Immunity and Cancer, University Paris, 75005 Paris, France; beatriz.grandalrejo@curie.fr (B.G.); elise.dumas@curie.fr (E.D.); eric.daoud@curie.fr (E.D.); floriane.jochum@curie.fr (F.J.); paul.gougis@curie.fr (P.G.); sph.michel@gmail.com (S.M.); enora.laas@curie.fr (E.L.); anne-sophie.hamy-petit@curie.fr (A.S.H.); 2Department of Surgical Oncology, Institut Curie, University Paris, 75005 Paris, France; ajaljehani@imamu.edu.sa; 3Department of Surgery, College of Medicine, Imam Mohammad Ibn Saud Islamic University (IMSIU), Riyadh 11564, Saudi Arabia; 4Department of Surgical Oncology, Institut Godinot, Université de Lorraine, 51100 Reims, France; judicael.hotton@reims.unicancer.fr; 5Department of Medical Oncology, Institut Godinot, Université de Lorraine, 51100 Reims, France; amelie.lemoine@reims.unicancer.fr; 6Henri Becquerel Cancer Center, Department of Pathology, INSERM U1245, UniRouen Normandy University, 76130 Rouen, France; marick.lae@chb.unicancer.fr; 7Department of Pathology, Institut Curie, University Paris, 75231 Paris, France; 8Department of Medical Oncology, Institut Curie, University Paris, 75231 Paris, France; jean-yves.pierga@curie.fr (J.-Y.P.); khaoula.alaouiismaili@curie.fr (K.A.I.); florence.lerebours@curie.fr (F.L.)

**Keywords:** breast cancer, neoadjuvant chemotherapy, season, seasonality, sunlight, response to treatment, pCR, immune infiltration, prognosis, survival

## Abstract

**Simple Summary:**

High tumor-infiltrating lymphocyte (TIL) levels are associated with an increased response to neoadjuvant chemotherapy (NAC) in breast cancer (BC). The seasonal fluctuation of TILs in breast cancer is poorly documented. In this study, we compared pre- and post-treatment immune infiltration, the treatment response as assessed by means of pathological complete response (pCR) rates, and survival according to the seasonality of BC diagnoses in a clinical cohort of patients treated with NAC. We found no association between seasonality and baseline TIL levels or pCR rates. We found that post-NAC stromal lymphocyte infiltration was lower when cancer was diagnosed in the summer, especially in the subgroup of patients with TNBC. Our data do not support the hypothesis that the seasonality of diagnoses has a major impact on the natural history of BC treated with NAC.

**Abstract:**

Breast cancer (BC) is the most common cancer in women worldwide. Neoadjuvant chemotherapy (NAC) makes it possible to monitor in vivo response to treatment. Several studies have investigated the impact of the seasons on the incidence and detection of BC, on tumor composition, and on the prognosis of BC. However, no evidence is available on their association with immune infiltration and the response to treatment. The objective of this study was to analyze pre- and post-NAC immune infiltration as assessed by TIL levels, the response to treatment as assessed by pathological complete response (pCR) rates, and oncological outcomes as assessed by relapse-free survival (RFS) or overall survival (OS) according to the seasonality of BC diagnoses in a clinical cohort of patients treated with neoadjuvant chemotherapy. Out of 1199 patients, the repartition of the season at BC diagnosis showed that 27.2% were diagnosed in fall, 25.4% in winter, 24% in spring, and 23.4% in summer. Baseline patient and tumor characteristics, including notable pre-NAC TIL levels, were not significantly different in terms of the season of BC diagnosis. Similarly, the pCR rates were not different. No association for oncological outcome was identified. Our data do not support the idea that the seasonality of diagnoses has a major impact on the natural history of BC treated with NAC.

## 1. Introduction

Breast cancer (BC) remains the commonest and most deadly cancer in women worldwide [1]. In 2020, approximately 2.3 million women in the world were diagnosed with breast cancer, and it caused 690,000 deaths [2]. Neoadjuvant or pre-operative chemotherapy (NAC) has historically been administrated to patients with inflammatory or locally advanced breast cancer (BC). Beyond increasing breast-conserving surgery rates [3], it also serves as an in vivo chemosensitivity test, helping to understand treatment resistance and facilitating the study of cancer biology [4]. Moreover, it aids in refining the prognoses of patients after NAC, as the response to treatment defined by the pathological complete response (pCR) is associated with substantially longer times to recurrence and death [3,5]. After NAC, pathological complete response (pCR) occurs in 30 to 50 percent of patients and is linked to prolonged relapse-free survival (RFS) and overall survival (OS) [6,7].

Over the past decade, tumor-infiltrating lymphocytes (TILs) have been extensively studied in BC [8,9,10,11]. Many studies have reported associations between high levels of TILs at diagnosis and better responses to and prognoses of neoadjuvant chemotherapy and adjuvant chemotherapy settings, particularly for triple-negative and *HER2*-positive breast carcinomas [12,13,14,15]. Emerging data suggest that TILs are associated with responses to both cytotoxic treatments and immunotherapy, particularly for patients with triple-negative BC [9].

Both intrinsic (spontaneous mutations) and extrinsic factors can impart cancer risk through the accumulation of DNA errors. Intrinsic risk factors contribute modestly (<10~30%) to cancer development. While several extrinsic risk factors have been identified for different cancers, no single one can account for their risk proportions, suggesting complex mechanisms for their etiologies. These factors include factors related to the characteristics of the host (gender, age, body mass index) [16,17,18,19,20], toxins (tobacco, alcohol consumption) [21,22], exogenous exposures (hormonal treatments, chronically used medications) [23,24], and lifestyle (nutritional factors, diet, physical activity, sleep duration) [25,26], as well as environmental factors, such as pollution [27], sunlight exposure, UV radiation, ionizing radiation [28], weather/temperature, and the seasons [29]. About 20% to 40% of cancer cases and almost half of cancer deaths can be potentially prevented through lifestyle and environmental changes [30,31,32]. It has long been observed for breast and prostate cancers in particular that extensive international geographical variations exist in their incidences [2]. Furthermore, immigrants moving from countries with lower cancer incidences to countries with higher cancer rates are soon subject to the higher risk in their new countries [33,34]. This adoption of the host-country incidence pattern is consistent with changes in factors present in each geographic region. Seasonal variations represent a well-known phenomenon for a number of noninfectious disorders, both in terms of incidence and mortality. Cardiovascular (coronary and cerebrovascular diseases) and respiratory diseases (pneumonia and influenza) are examples of diseases that contribute to the greater number of winter deaths [35,36,37]. Seasons are associated with variations in temperature, sunlight, UV, and vitamin D exposure, as well as food intake. Such changes may affect human physiology [38], metabolism [39], circulation, levels of inflammation [40], and hormonal secretions, such as estrogen or melatonin [41,42]. Food intake [43], such as meat consumption [39] and fruit and vegetable intake [44,45], also varies throughout the year.

In oncology, seasonality is a topic that has raised little interest to date. Several studies have reported seasonal variations in the detection of BC, with an increased frequency of tumor detection in fall and winter [46,47]. Conversely, others have not seen such patterns [48]. Some authors have reported differences in cancer biology and the composition of tumors throughout the year [49,50,51]. Finally, the prognostic impact of seasons on relapse and mortality has also been brought up by several authors [29,52,53,54], with there being an overall protective effect for tumors with summer diagnoses rather than winter diagnoses. However, such associations have not been extensively described in the neoadjuvant setting, and up-to-date evidence regarding real-world seasonal variations in the natural history of BC is lacking in this specific context.

The current study aimed to analyze pre- and post-NAC immune infiltration as assessed by TIL levels, the response to treatment as assessed by pathological complete response (pCR) rates, and oncological outcomes as assessed by relapse-free survival and overall survival according to seasonality in a clinical cohort of patients treated with neoadjuvant chemotherapy.

## 2. Materials and Methods

### 2.1. Patients and Tumors

The analysis was performed on 1199 patients with invasive local or locally advanced breast cancer (stage T1–T3NxM0) treated with NAC at the Institut Curie, Paris, between January 2002 and April 2012. The cohort included unifocal, unilateral, and primary tumors, excluding T4 or metastatic tumors. All patients were treated with NAC, and additional treatments were decided in accordance with national guidelines. Four to six weeks following the conclusion of treatment, surgery was undertaken. The examination of retrospective data was approved by the Breast Cancer Study Group of the Institut Curie (CNIL declaration number 1547270, NEOREP Cohort). The study of tissue specimens and patients was carried out following institutional and ethical guidelines. Patient written informed consent was not necessary under French rules.

### 2.2. Tumor Samples and BC Subtype

Two pathologists with expertise in cancer pathology examined the tumor samples (DdC, ML). Prior to therapy, a core needle biopsy (CNB) was used to confirm the pathological diagnosis. Cases were designated ER- or PR-negative according to French national criteria if less than 10% of tumor cells expressed ER/PR [55]. Immunohistochemistry (IHC) was used to evaluate HER2 expression, with grading based on the American Society of Clinical Oncology (ASCO)/College of American Pathologists (CAP) criteria [56]. BC subtypes were defined as follows: tumors positive for either ER or PR and negative for HER2 were classified as luminal; tumors positive for HER2 were considered HER2-positive BC; tumors negative for ER, PR, and HER2 were considered as triple-negative BC (TNBC).

### 2.3. Tumor-Infiltrating Lymphocytes

Tumor-infiltrating lymphocytes (TIL) levels were assessed on formalin-fixed paraffin-embedded (FFPE) tumor tissue samples from pretreatment core needle biopsies and the corresponding post-NAC surgical specimens, following the recommendations of the international TILs Working Group, before [11] and after NAC [8]. TILs were reviewed between January 2015 and March 2017 for research purposes and were defined as the presence of a mononuclear cell infiltrate (including lymphocytes and plasma cells and excluding polymorphonuclear leukocytes) [15]. TILs in direct contact with tumor cells were counted as intra-tumoral TILs (IT TILs) and those in the peri-tumoral areas as stromal TILs (str TILs). They were evaluated both in the stroma and within the tumor scar border, after excluding areas around ductal carcinoma in situ, tumor zones with necrosis, and artifacts. TIL levels were scored continuously as the average percentage of stroma area occupied by mononuclear cells, in deciles, and in binary, with a cut-off of 30% to divide patients into TIL high and low groups [13].

### 2.4. Response to Treatment

We defined pathological complete response (pCR) as the absence of invasive residual tumors from both the breast and axillary nodes (ypT0/is N0) after neoadjuvant chemotherapy [57].

### 2.5. Date and Season of BC Diagnosis

The date of breast cancer diagnosis was defined as the date of the first core biopsy with cancer. If the biopsy date was not available, the date of the first physical examination and then the date of the first breast imaging were used. In addition, the dates of the breast cancer diagnoses were grouped into four seasons following a methodology described previously [46]: spring (from 1 March to 31 May); summer (from 1 June to 31 August); fall (from 1 September to 30 November), and winter (from 1 December to 28–29 February). Breast cancer diagnosis dates were grouped by month and year. We considered the year to start in December and end in November to facilitate the data visualization. We considered all patients to belong to the same type of temperate climate, given the latitude and surface of metropolitan France, as well as the absence of dispersion in our patients.

### 2.6. Survival Endpoints

The period from surgery to death, locoregional recurrence, or distant recurrence, whichever came first, was described as the relapse-free survival (RFS). Distant recurrence-free survival (DRFS) was defined as the time from operation to the first distant recurrence or death, while overall survival (OS) was defined as the time from surgery to death. Patients that did not show any of these occurrences documented were censored at the last known contact date. The survival analysis cut-off date was 1 February 2019.

### 2.7. Statistical Analysis

For qualitative variables, the study population was expressed in terms of frequencies, whereas medians and corresponding ranges were used for quantitative variables. For each variable, Chi-squared tests were used to explore differences between subgroups (*p*-values lower than 0.05 were deemed significant). For groups of less than 30 patients and variables with multimodal distributions, Wilcoxon–Mann–Whitney tests were used to compare continuous variables across groups. In all other situations, Student’s *t*-test was utilized. The Kaplan–Meier approach was used to estimate survival probability, and log-rank tests were used to compare survival curves. The Cox proportional hazards model was used to determine hazard ratios and associated 95 percent confidence intervals. In the univariate analysis, variables with a *p*-value for the likelihood ratio test of 0.05 or below were chosen for inclusion in the multivariate analysis. The final multivariate model was built using a forward stepwise selection approach with a significance criterion of 5%. R software version 4.0.3 (Initially written by Robert Gentleman and Ross Ihaka, Auckland, New Zealand, www.cran.r-project.org, accessed on 10 October 2020) was used to process data and perform statistical analyses.

## 3. Results

### 3.1. Study Population and Tumors Characteristics

One thousand one hundred ninety-nine patients were treated with NAC and included in our cohort (Table 1). The date of breast cancer diagnosis was available in all cases. In total, 528 patients (44.0%) had luminal breast cancer, 376 patients had triple-negative breast cancer (TNBC) (31.4%), and 295 (24.6%) had *HER2*-positive BC. The median age was 48.6 years (range: 23.6–79.5). Most of the patients were premenopausal (n = 747, 62.8%) and had a normal BMI (n = 681, 57.1%). For 69.7% of the tumors, the diagnosis was made at the T2 stage (n = 62), mostly with baseline axillary node involvement (n = 51, 57.3%).

### 3.2. Seasonality of BC Diagnosis

The patient allocations were not evenly distributed over all the years, as not all the files for the first (2002) and last (2012) years of recruitment were collected. There was a progressive increase in patients treated with neoadjuvant chemotherapy from 2003 to 2011. Significant seasonal variations were found in the diagnosis of breast cancer. The number of cases tended to peak in the cold seasons (winter or fall), except for the years 2006 and 2007 (Figure 1). Patients and tumor characteristics did not differ by season at BC diagnosis, and this was true both in the whole population (Table 1) and after subgroup analysis by BC subtype (Appendix A). Baseline pre-NAC TIL levels were available in 717 patients and did not significantly differ according to the season at BC diagnosis. This was true for str TIL levels in the whole population (Figure 2A) and in each BC subtype (Figure 2B), irrespective of the cut-off for TILs analysis (Figure 2C,D), as well as for IT TIL levels (Figure 2E,H). The same results were observed in the analysis by month (str TIL levels, *p* = 0.39; IT TIL levels, *p* = 0.19) (Appendix A).

### 3.3. Response to Treatment and Post-NAC TIL Levels

PCR rates did not significantly differ according to seasonality (*p* = 0.87), as was the case for each BC subtype (Figure 3, Appendix A) and the different chemotherapy regimens (Appendix A), nor were there any associations between seasonality and chemotherapy resistance (*p* = 0.13, Appendix A). Conversely, post-NAC str TILs levels differed significantly between seasons (*p* = 0.01). Post-NAC str TILs levels were significantly lower in patients diagnosed with BC during the summer and spring compared to patients diagnosed in the winter (both, *p* = 0.01). After performing a subgroup analysis of breast cancer subtypes, seasonal variation and month of diagnosis were found to be statistically significant only in TNBC (*p* = 0.04 and *p* = 0.06, respectively) (Figure 4 and Appendix A, Table 1 and Appendix A). Tumors diagnosed in January had a significantly higher stromal infiltrate than for patients diagnosed in June (*adjusted-p* = 0.02, Appendix A).

### 3.4. Survival Analysis

Seasonality of BC diagnoses was not significantly associated with RFS, DRFS, or OS in the whole population (Figure 5A–H, Appendix A). However, there was a trend towards significance in the group of patients with the luminal BC subtype (*p* = 0.06 for RFS). Therefore, a univariate cox analysis was performed for the luminal BC group. Patients with luminal BC diagnosed in winter had worse prognoses than those diagnosed in summer (HR 0.55 (95% CI, 0.36–0.86), *p* = 0.01).

## 4. Discussion

In this study analyzing immune infiltration, response to treatment, and oncologic outcomes according to the seasonality of BC diagnosis in a cohort of patients treated with NAC, we found little, if any, association between seasonality and the parameters we analyzed. Our study adds insights to the existing evidence, summarized in Appendix A.

***First***, in our cohort, the season of diagnosis was evenly distributed throughout the year. In contrast, out of 2,921,714 BC cases diagnosed worldwide, Oh and colleagues [58] found that BC was consistently diagnosed more often in spring and fall, and irrespective of menopausal status. Among 2895 patients who self-detected an incident BC, Ross et al. [59] found that the monthly peaks in detection occurred in spring and late fall, though these seasonal variations were significant only for ER-negative tumors but not for ER-positive ones. The Swedish cancer register also identified a clear decrease in the frequency of new cases diagnosed during the summer months, followed by a transient increase in subsequent months [47]. Seasonal variations with winter peaks were also evidenced in Korea [46]. The authors concluded that accessibility to healthcare services and logistics for screening programs are two of the most significant factors affecting the seasonality of cancer detection. The lack of an association between season of diagnosis and incidence in our study could be explained by the characteristics of the population, consisting of patients with large tumors treated with NAC, which are classically not diagnosed within screening programs but are rather interval-diagnosed tumors [60].

***Second***, we did not find any differences regarding any of the tumor characteristics at diagnosis or in the baseline immune infiltration, and this finding was true also for the subgroup analysis subset by BC subtype. This finding is interesting because it has been suggested that social factors potentially modifying the utilization patterns of healthcare programs, such as holidays, religious events, or the organization of health care, can notably modify tumor size or clinical presentation at diagnosis. Out of 905 patients diagnosed with BC, Paradiso et al. found that tumor size and nodal involvement did not differ in the different seasons of the year. In this study, ER and PR showed significant periodicities, with peaks in January/April and in July, respectively [51]. Though the timing of the peak for PR seasonality remains under debate, several publications have also reported circannual variations in PR [49,51]. Studying 738 tumors, Joensuu and colleagues found that the amount of tumor necrosis was associated with the months of the diagnoses [50]. Regarding baseline immune infiltration, previous work outside the field of cancer studies has shown that seasonal variations in immunity appear to occur in humans [61], but we did not identify such patterns for the levels of immune infiltration in breast tumors.

***Third***, response to NAC was not modified by seasonality. To our knowledge, we report data on response to chemotherapy for the first time. Such a negative association can be explained by either: (i) a genuine absence of effect or (ii) a lack of power below the level needed to detect a slight effect. It would be of interested for other studies to investigate whether seasonality impacts the response to chemotherapy or to the immunotherapies that are becoming cornerstones of other cancer localizations, such as lung cancer and melanoma.

***Fourth***, in our cohort, post-NAC stromal TILs were lower when the BC was diagnosed in summer compared to the other seasons. Our team previously showed that post-NAC TILs are very low and mainly driven by response to treatment [15] and by the “pCR” or “no pCR” status of the tumor after NAC. We previously failed to demonstrate any association with response to NAC in the cohort; as such, we cannot exclude the possibility that this association could be an incidental finding.

***Fifth***, we did not find any association with oncologic outcomes. The literature on the impact of seasonality on mortality shows discordant findings. Several small-sized studies found a positive prognostic impact from BC diagnosis in the summer [29,54,62,63], others found no impact [64,65], while the largest study performed so far, utilizing 89,630 BC cases from Swedish registers, found an increased risk of death in women diagnosed in the summer [66]. For the neoadjuvant setting, our data suggest that the impact of seasonality, if it exists, would not be major.

Overall, our study fills a gap in the existing literature on seasonality by describing the first cohort including BC patients treated with NAC. The study limitations include constraints on the research design and methodology, as well as the low efficacy, potentially leading to a lack of statistical power. Findings from one country or a given health care system may not be generalizable to other countries and sociological settings. The role of seasonality in natural history is complex and multifactorial, and we cannot exclude the possibility that a larger number of patients together with more detailed sociodemographic, clinical, and biological parameters could have unmasked significant associations. It was not possible to assess the physiological levels of vitamin D or melatonin as proxies for interannual and inter-regional variations in sunshine hours, temperature, and rainfall and the limited but existing changes in climate throughout Metropolitan France. Finally, we did not perform immunophenotyping to correct the differentiation in the tumor immune profiling. Recent data suggest that, while high TIL infiltration is associated with better outcomes, in ER-positive breast cancer patients, only immune infiltrates expressing PD-1+ and exhausted CD8+ T cells can predict the response to immune checkpoint inhibitors [67]. However, there is currently no clear consensus on which immunophenotyping panel or antibody combination should be used in breast cancer, and their clinical interpretations are not standardized [68]. One advantage of quantitative TIL assessment is that it can be performed routinely in any pathology department without increasing technical costs. Furthermore, some studies have reported an unequivocal correlation between the numbers of unstained TILs and CD8+ TILs [15], CD3 counts, and counts for other immune subpopulations (CD3+, CD20+, CD68+) [69]. This relation supports the notion that quantitative assessments could serve as relevant surrogate markers. However, from a research standpoint, extensive characterization of the immune phenotype to determine the subsets of TILs present could be of interest. Until further associations are validated, seasonality should not be considered a key element to tailor BC diagnosis, treatments, and follow-up interventions.

## Figures and Tables

**Figure 1 cancers-14-03080-f001:**
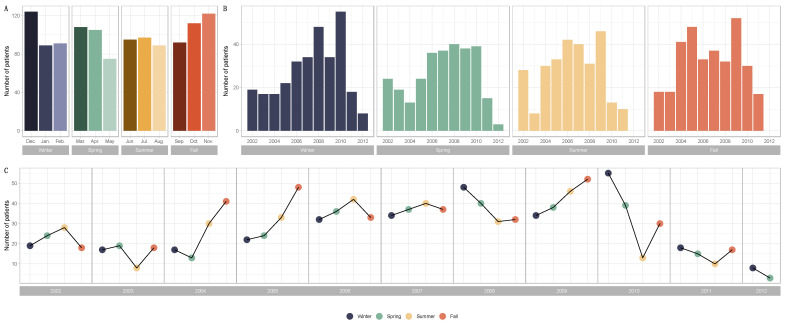
Barplots and line graphs for numbers of patients by season and over the years (Winter, n = 304; Spring, n = 288; Summer, n = 281; Fall, n = 326) and over the years (2002, n = 89; 2003, n = 62; 2004, n = 101; 2005, n = 127; 2006, n = 143; 2007, n = 148; 2008, n = 151; 2009, n = 170; 2010, n = 137; 2011, n = 60; 2012, n = 11). (**A**) Barplots for the distribution of breast cancer diagnoses by month and season. (**B**) Barplots for the distribution of breast cancer diagnoses by year and season. (**C**) Line graphs for breast cancer diagnoses by year and season.

**Figure 2 cancers-14-03080-f002:**
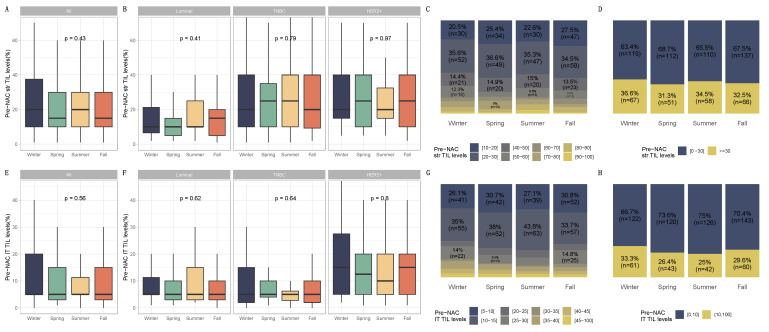
Pre-NAC TILs and season at breast cancer diagnosis in the general population and by breast cancer subtype. BC patients with pre-NAC str TIL levels available [n = 717] and IT TIL levels [717]. The first and third quartiles are represented by the bottom and top bars of the boxplots, respectively; the median is represented by the medium bar; and whiskers extend to 1.5 times the interquartile range. (**A**) Stromal lymphocytes among the whole population. (**B**) Stromal lymphocytes in each BC subtype. (**C**) Pre-NAC stromal lymphocyte counts were binned by 10% increments in patients by season to determine the proportion of tumor. (**D**) Percentage of tumor according to pre-NAC stromal lymphocyte levels binned by Denkert cut-off by season. (**E**) Intratumoral lymphocytes among the whole population. (**F**) Intratumoral lymphocytes in each BC subtype. (**G**) Pre-NAC intratumoral lymphocyte counts were binned by 10% increments in patients by season to determine the proportion of tumor. (**H**) Percentage of tumor according to pre-NAC intratumoral lymphocyte levels binned by Denkert cut-off by season.

**Figure 3 cancers-14-03080-f003:**
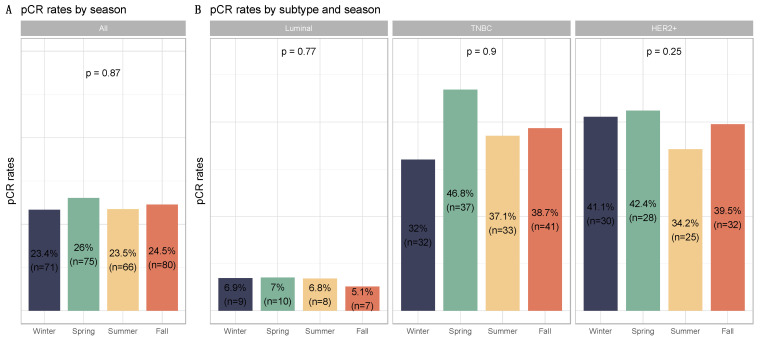
Barplots for associations between response to treatment and season at breast cancer diagnosis in the whole population and by breast cancer subtype: (**A**) among the whole population, All (n = 1193) (Winter (n = 303), Spring (n = 287), Summer (n = 279), Fall (n = 324)); (**B**) by BC subtype, Luminal (n = 526) (Winter (n = 103), Spring (n = 142), Summer (n = 117), Fall (n = 137)); TNBC (n = 374) (Winter (n = 100), Spring (n = 79), Summer (n = 89), Fall (n = 106)); HER2 (n = 293) (Winter (n = 73), Spring (n = 66), Summer (n = 73), Fall (n = 81)).

**Figure 4 cancers-14-03080-f004:**
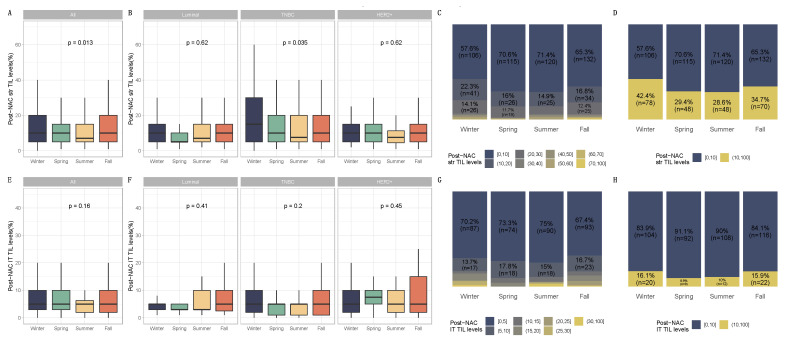
Post-NAC TIL levels by season at BC diagnosis in the general population and by breast cancer subtype. BC patients with post-NAC str TIL levels available [n = 717] and IT TIL levels [483]. The first and third quartiles are represented by the bottom and top bars of the boxplots, respectively; the median is represented by the medium bar, and the whiskers extend to 1.5 times the interquartile range. (**A**) Stromal lymphocytes among the whole population. (**B**) Stromal lymphocytes in each BC subtype. (**C**) Percentage of tumor based on post-NAC stromal lymphocyte counts in patients binned by 10% increments by season. (**D**) Percentage of tumor according to post-NAC stromal lymphocyte levels binned by Denkert cut-off by season. (**E**) Intratumoral lymphocytes among the whole population. (**F**) Intratumoral lymphocytes in each BC subtype. (**G**) Percentage of tumor based on post-NAC intratumoral lymphocyte counts in patients binned by 10% increments by season. (**H**) Percentage of tumor according to post-NAC intratumoral lymphocytes level binned by Denkert cut-off by season.

**Figure 5 cancers-14-03080-f005:**
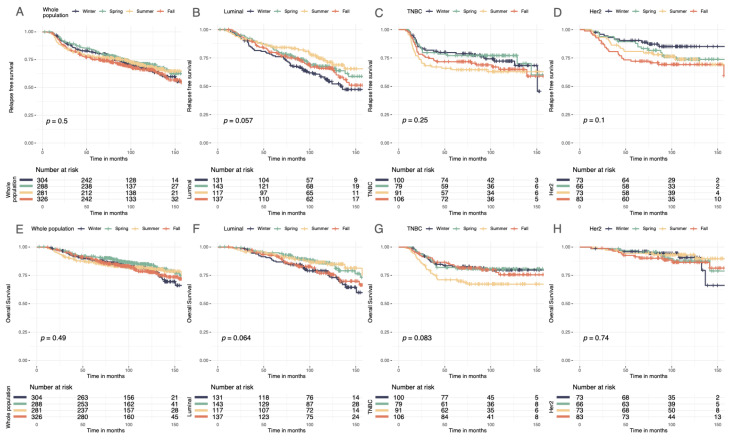
Relapse-free and overall Kaplan–Meier survival curves by season at BC diagnosis. (**A**) Relapse-free survival curves according to season. (**B**) Relapse-free survival curves in luminal breast cancer according to season. (**C**) Relapse-free survival curves in TNBC breast cancer according to season. (**D**) Relapse-free survival curves in *Her2*-positive breast cancer according to season. (**E**) Overall survival curves according to season. (**F**) Overall survival curves in luminal breast cancer according to season. (**G**) Overall survival curves in TNBC breast cancer according to season. (**H**) Overall survival curves in *Her2*-positive breast cancer according to season.

**Table 1 cancers-14-03080-t001:** Patients and tumor characteristics among the whole population and according to season.

	Class	All	Winter	Spring	Summer	Fall	*p*	test
	n	1199	304	288	281	326		

Age at BC diagnosis		48.20 [40.90, 55.50]	46.95 [39.90, 53.92]	47.75 [41.80, 54.65]	49.80 [41.50, 57.30]	47.70 [41.90, 55.90]	0.175	nonnorm
Menopausal status	Premenopausal	747 (62.8)	200 (66.2)	182 (64.1)	166 (59.1)	199 (61.8)	0.318	
	Postmenopausal	442 (37.2)	102 (33.8)	102 (35.9)	115 (40.9)	123 (38.2)		
BMI	18.5–24.9	681 (57.1)	173 (57.3)	171 (59.6)	156 (55.9)	181 (55.7)	0.699	
	<18.5	48 (4.0)	9 (3.0)	10 (3.5)	12 (4.3)	17 (5.2)		
	25–29.9	304 (25.5)	82 (27.2)	72 (25.1)	75 (26.9)	75 (23.1)		
	>=30	160 (13.4)	38 (12.6)	34 (11.8)	36 (12.9)	52 (16.0)		
BMI		23.80 [21.50, 27.30]	24.05 [21.50, 27.20]	23.50 [21.50, 26.80]	24.10 [21.70, 27.15]	23.50 [21.20, 27.70]	0.768	nonnorm
Smoking status	Never	623 (65.2)	175 (69.2)	142 (62.8)	141 (64.7)	165 (63.7)	0.476	
	Current	179 (18.7)	48 (19.0)	43 (19.0)	37 (17.0)	51 (19.7)		
	Former	154 (16.1)	30 (11.9)	41 (18.1)	40 (18.3)	43 (16.6)		
Year BC diagnosis	2002	89 (7.4)	19 (6.2)	24 (8.3)	28 (10.0)	18 (5.5)	**<0.001**	
	2003	62 (5.2)	17 (5.6)	19 (6.6)	8 (2.8)	18 (5.5)		
	2004	101 (8.4)	17 (5.6)	13 (4.5)	30 (10.7)	41 (12.6)		
	2005	127 (10.6)	22 (7.2)	24 (8.3)	33 (11.7)	48 (14.7)		
	2006	143 (11.9)	32 (10.5)	36 (12.5)	42 (14.9)	33 (10.1)		
	2007	148 (12.3)	34 (11.2)	37 (12.8)	40 (14.2)	37 (11.3)		
	2008	151 (12.6)	48 (15.8)	40 (13.9)	31 (11.0)	32 (9.8)		
	2009	170 (14.2)	34 (11.2)	38 (13.2)	46 (16.4)	52 (16.0)		
	2010	137 (11.4)	55 (18.1)	39 (13.5)	13 (4.6)	30 (9.2)		
	2011	60 (5.0)	18 (5.9)	15 (5.2)	10 (3.6)	17 (5.2)		
	2012	11 (0.9)	8 (2.6)	3 (1.0)	0 (0.0)	0 (0.0)		
Genetic variants	No	221 (82.8)	58 (78.4)	55 (85.9)	48 (81.4)	60 (85.7)	0.581	
	Yes	46 (17.2)	16 (21.6)	9 (14.1)	11 (18.6)	10 (14.3)		
Clinical Tumor size (mm)		40.00 [30.00, 55.00]	40.00 [30.00, 55.00]	40.00 [30.00, 50.00]	40.00 [35.00, 55.00]	45.00 [35.00, 60.00]	0.332	nonnorm
Clinical T stage (TNM)	T0-T1	70 (5.8)	19 (6.2)	13 (4.5)	20 (7.1)	18 (5.5)	0.372	
	T2	798 (66.6)	201 (66.1)	207 (72.1)	181 (64.4)	209 (64.1)		
	T3-T4	330 (27.5)	84 (27.6)	67 (23.3)	80 (28.5)	99 (30.4)		
Clinical N stage (TNM)	N0	525 (43.8)	120 (39.5)	141 (49.0)	130 (46.4)	134 (41.1)	0.067	
	N1-N2-N3	673 (56.2)	184 (60.5)	147 (51.0)	150 (53.6)	192 (58.9)		
SBR grade	Grade I	47 (4.1)	11 (3.7)	17 (6.0)	13 (4.9)	6 (1.9)	0.249	
	Grade II	432 (37.3)	107 (36.1)	104 (37.0)	104 (39.0)	117 (37.4)		
	Grade III	678 (58.6)	178 (60.1)	160 (56.9)	150 (56.2)	190 (60.7)		
KI67		30.00 [16.00, 55.00]	27.50 [16.25, 50.00]	30.00 [15.00, 50.00]	30.00 [15.00, 55.00]	33.00 [18.00, 60.00]	0.438	nonnorm
KI67	[0–10)	65 (11.2)	12 (8.5)	20 (13.8)	20 (14.6)	13 (8.2)	0.479	
	[10–20)	110 (18.9)	30 (21.1)	27 (18.6)	24 (17.5)	29 (18.4)		
	>=20	407 (69.9)	100 (70.4)	98 (67.6)	93 (67.9)	116 (73.4)		
Mitotic index		15.00 [7.00, 28.00]	14.00 [7.25, 28.00]	14.00 [6.00, 28.00]	14.00 [6.75, 26.00]	15.00 [7.00, 30.00]	0.967	nonnorm
Mitotic index	[0–7) mitose/2 mm2	341 (31.5)	80 (29.6)	89 (33.2)	76 (31.1)	96 (32.1)	0.617	
	[7–13) mitose/2 mm2	295 (27.3)	81 (30.0)	64 (23.9)	74 (30.3)	76 (25.4)		
	>=13 mitose ou plus/2 mm2.	445 (41.2)	109 (40.4)	115 (42.9)	94 (38.5)	127 (42.5)		
BC subtype	Luminal	528 (44.0)	131 (43.1)	143 (49.7)	117 (41.6)	137 (42.0)	0.499	
	TNBC	376 (31.4)	100 (32.9)	79 (27.4)	91 (32.4)	106 (32.5)		
	HER2+	295 (24.6)	73 (24.0)	66 (22.9)	73 (26.0)	83 (25.5)		
DCIS component	No	604 (60.8)	159 (61.9)	130 (53.9)	142 (60.9)	173 (66.0)	0.048	
	Yes	389 (39.2)	98 (38.1)	111 (46.1)	91 (39.1)	89 (34.0)		
Stromal TIL levels (%)		20.00 [10.00, 30.00]	20.00 [10.00, 37.50]	15.00 [10.00, 30.00]	20.00 [10.00, 30.00]	15.00 [10.00, 30.00]	0.425	nonnorm
Stromal TIL levels (%)	[0–30]	475 (66.2)	116 (63.4)	112 (68.7)	110 (65.5)	137 (67.5)	0.730	
	>=30	242 (33.8)	67 (36.6)	51 (31.3)	58 (34.5)	66 (32.5)		
IT TIL levels (%)		5.00 [5.00, 15.00]	5.00 [5.00, 20.00]	5.00 [3.00, 15.00]	5.00 [5.00, 11.25]	5.00 [3.00, 15.00]	0.559	nonnorm
IT TIL levels (%)	[0, 10]	511 (71.3)	122 (66.7)	120 (73.6)	126 (75.0)	143 (70.4)	0.315	
	(10, 100]	206 (28.7)	61 (33.3)	43 (26.4)	42 (25.0)	60 (29.6)		
LVI	No	267 (61.0)	61 (59.2)	69 (63.3)	69 (60.5)	68 (60.7)	0.942	
	Yes	171 (39.0)	42 (40.8)	40 (36.7)	45 (39.5)	44 (39.3)		
Histological type	NST	1062 (93.5)	276 (94.5)	256 (93.1)	247 (92.9)	283 (93.4)	0.859	
	Others	74 (6.5)	16 (5.5)	19 (6.9)	19 (7.1)	20 (6.6)		

Pathological complete response	No	901 (75.5)	232 (76.6)	212 (73.9)	213 (76.3)	244 (75.3)	0.870	
	Yes	292 (24.5)	71 (23.4)	75 (26.1)	66 (23.7)	80 (24.7)		
RCB index (continuous)		1.82 [0.00, 3.06]	1.76 [0.00, 2.77]	1.67 [0.00, 3.01]	1.79 [0.00, 3.04]	2.06 [0.00, 3.30]	0.203	nonnorm
Residual Cancer Burden class	RCB-0	202 (28.2)	47 (25.7)	54 (33.1)	45 (26.8)	56 (27.6)	0.126	
	RCB-I	65 (9.1)	24 (13.1)	11 (6.7)	15 (8.9)	15 (7.4)		
	RCB-II	309 (43.1)	84 (45.9)	68 (41.7)	77 (45.8)	80 (39.4)		
	RCB-III	141 (19.7)	28 (15.3)	30 (18.4)	31 (18.5)	52 (25.6)		
ypN	0	682 (57.0)	172 (56.8)	162 (56.4)	173 (61.6)	175 (53.7)	0.746	
	[1–3]	341 (28.5)	84 (27.7)	83 (28.9)	68 (24.2)	106 (32.5)		
	[4–9]	145 (12.1)	40 (13.2)	35 (12.2)	33 (11.7)	37 (11.3)		
	10 and more	29 (2.4)	7 (2.3)	7 (2.4)	7 (2.5)	8 (2.5)		
Stromal TIL levels (%) (post-NAC)		10.00 [5.00, 15.00]	10.00 [5.00, 20.00]	10.00 [5.00, 15.00]	7.00 [5.00, 15.00]	10.00 [5.00, 20.00]	**0.013**	nonnorm
Stromal TIL levels (%) (post-NAC)	[0, 10]	473 (66.0)	106 (57.6)	115 (70.6)	120 (71.4)	132 (65.3)	**0.023**	
	(10, 100]	244 (34.0)	78 (42.4)	48 (29.4)	48 (28.6)	70 (34.7)		
IT TIL levels (%) (post-NAC)		5.00 [2.00, 10.00]	5.00 [3.00, 10.00]	5.00 [3.00, 10.00]	5.00 [2.00, 6.25]	5.00 [2.00, 10.00]	0.163	nonnorm
IT TIL levels (%) (post-NAC)	[0, 10]	420 (87.0)	104 (83.9)	92 (91.1)	108 (90.0)	116 (84.1)	0.207	
	[10, 100]	63 (13.0)	20 (16.1)	9 (8.9)	12 (10.0)	22 (15.9)		

*Missing data:* menopausal status, n = 10; BMI, n = 6; smoking status, n = 243; hereditary predisposition, n = 932; clinical tumor size (mm), n = 1; clinical T stage (TNM), n = 1; clinical N stage (TNM), n = 1; SBR grade, n = 42; KI67, n = 617; mitotic index, n = 118; DCIS component, n = 206; stromal TIL levels (%), n = 482; IT TIL levels (%), n = 482; LVI, n = 761; histological type, n = 63; pathological complete response, n = 6; RCB index (continuous), n = 482; residual cancer burden class, n = 482; ypN, n = 2; stromal TIL levels (%) (post-NAC), n = 482; IT TIL levels (%) (post-NAC), n = 716. *Abbreviations:* NAC = neoadjuvant chemotherapy; BMI = body mass index; TNBC = triple-negative breast cancer; str TILs = stromal tumor-infiltrating lymphocytes; IT TILs = intratumoral-infiltrating lymphocytes; pCR = pathologic complete response; RCB = residual cancer burden. “n” denotes the number of patients. In cases of categorical variables, percentages are expressed in brackets. In cases of continuous variables, the mean value is reported, with the standard deviation in brackets. In cases of nonnormal continuous variables, the median value is reported, with the interquartile range in brackets.

## Data Availability

Data are available on request due to privacy/ethical restrictions. The data that support the findings of this study are available on request from the corresponding author (FR). The data are not publicly available because they contain information that could compromise the privacy of the research participants.

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
