# Peer review of "No Impact of Seasonality of Diagnoses on Baseline Tumor Immune Infiltration, Response to Treatment, and Prognosis in BC Patients Treated with NAC"

_cancers, 2022, doi:10.3390/cancers14133080_

Round 1

Reviewer 1 Report

Line 68 Multiple extrinsic factors are listed which may modify the natural history of breast cancer. It would be helpful to provide some references for this statement.

Methods. Is the Institut Curie, Paris, a tertiary referral center including patients throughout France. If so, is there additional variability in the nature of seasons between regions of France, as there would be, say between Minnesota and Florida in the winter in the United States?.

Table 1. Does hereditary predisposition refer to mutational status, or simply family history?

Line 107. Considerable personal information is provided about these subjects in Table 1, and it would be helpful to know what the “French Rules” are that preclude requiring an informed consent.

Line 259, Fig 5, luminal. It is stated that patients diagnosed with luminal BCs in winter have a worse prognosis than those diagnosed in summer (HR 0.55 [95% CI, 0.36- 0.86], p= 0.009). However, te figure indicates these differences were not statistically significant (p = 0.057). Was this a subgroup analysis.

The limitations of the study should be discussed.

Abstract and discussion: it is stated in the abstract that Post-NAC TIL levels were lower when BC was diagnosed in the summer than in winter and fall (p=0.028), and this association was limited to the subset of patients with TNBC.. The placement of this statement in the abstract would indicate the authors consider this to be an important finding, even though it may be an incidental finding.  The potential clinical significance of this should be discussed.

Reviewer 2 Report

The manuscript investigates the potential link between seasonality and response to neoadjuvant chemotherapy and levels of tumor infiltrating lymphocytes. While no association was identified, the lack of association is interesting and clinically relevant. The study analyzed 1,199 women with T1-T3M0 breast cancers.  There was an interesting seasonal variation in ER and PR. The study is straightforward and well executed.

I have one concern- just because lymphocytes are present, it doesn't mean these lymphocytes are functional. The manuscript would be greatly strengthened by analysis of T-cell exhaustion markers.  I realize this is a large undertaking - perhaps a subset of samples could be analyzed.

Reviewer 3 Report

In this manuscript, the authors provide evidence indicating no association between sensonal factors and breast cancer incidence/response to neoadjuvant therapy and other clinical features. The statistical analyses are valid.

The findings are interesting. I will suggest the authors elaborate more on the reasons for the study to highlight the importance of their findings. I believe pretty many readers will have assumed that clinical features of breast cancer should not associate with seasonality.  

The authors still have to clarify some issues:

1.       Regarding neoadjuvant therapy, what kind of chemo-drug was used? Do different drugs show similar results?

2.       How about the risk of distant metastasis and drug resistance? 

Round 2

Reviewer 2 Report

The reviewers have adequately addressed concerns.